# Phytochemical Composition, Antioxidant Activity, and Enzyme Inhibitory Activities (α-Glucosidase, Xanthine Oxidase, and Acetylcholinesterase) of *Musella lasiocarpa*

**DOI:** 10.3390/molecules26154472

**Published:** 2021-07-24

**Authors:** Rurui Li, Yuerong Ru, Zhenxing Wang, Xiahong He, Kin-Weng Kong, Tingting Zheng, Xuechun Zhang

**Affiliations:** 1Key Laboratory for Forest Resources Conservation and Utilization in the Southwest Mountains of China, Ministry of Education, Southwest Forestry University, Kunming 650224, China; lirurui2021@163.com (R.L.); wangzhenxingfood@163.com (Z.W.); 2College of Life Science, Southwest Forestry University, Kunming 650224, China; ruyue13732704447@163.com; 3College of Horticulture and Landscape, Southwest Forestry University, Kunming 650224, China; 4Department of Molecular Medicine, Faculty of Medicine, University of Malaya, Kuala Lumpur 50603, Malaysia; kongkm@um.edu.my; 5Key Laboratory of Leather Chemistry and Engineering, Ministry of Education, Sichuan University, Chengdu 610065, China; zhengtingtings@163.com

**Keywords:** *Musella lasiocarpa*, antioxidant activity, α-glucosidase, xanthine oxidase, acetylcholinesterase, HPLC-MS/MS, GC-MS

## Abstract

In this study, we aimed to investigate the chemical components and biological activities of *Musella lasiocarpa*, a special flower that is edible and has functional properties. The crude methanol extract and its four fractions (petroleum ether, ethyl acetate, n-butanol, and aqueous fractions) were tested for their total antioxidant capacity, followed by their α-glucosidase, acetylcholinesterase, and xanthine oxidase inhibitory activities. Among the samples, the highest total phenolic and total flavonoid contents were found in the ethyl acetate (EtOAc) fraction (224.99 mg GAE/g DE) and crude methanol extract (187.81 mg QE/g DE), respectively. The EtOAc fraction of *Musella lasiocarpa* exhibited the strongest DPPH· scavenging ability, ABTS·^+^ scavenging ability, and α-glucosidase inhibitory activity with the IC_50_ values of 22.17, 12.10, and 125.66 μg/mL, respectively. The EtOAc fraction also showed the strongest ferric reducing antioxidant power (1513.89 mg FeSO_4_/g DE) and oxygen radical absorbance capacity ability (524.11 mg Trolox/g DE), which were higher than those of the control BHT. In contrast, the aqueous fraction demonstrated the highest acetylcholinesterase inhibitory activity (IC_50_ = 10.11 μg/mL), and the best xanthine oxidase inhibitory ability (IC_50_ = 5.23 μg/mL) was observed from the crude methanol extract as compared with allopurinol (24.85 μg/mL). The HPLC-MS/MS and GC-MS analyses further revealed an impressive arsenal of compounds, including phenolic acids, fatty acids, esters, terpenoids, and flavonoids, in the most biologically active EtOAc fraction. Taken together, this is the first report indicating the potential of *Musella lasiocarpa* as an excellent natural source of antioxidants with possible therapeutic, nutraceutical, and functional food applications.

## 1. Introduction

*Musella lasiocarpa* (Franch.) belongs to the monotypic genus *Musella* in the family *Musaceae* that is found primarily in southwestern China. It has attracted increasing attention around the world for its lotus-like golden inflorescence with more than eight-month anthesis [1,2,3]. *M. lasiocarpa* is traditionally used as medicine, food, and fodder [4]. It is a rich source of nutrients, and its comprehensive nutritional value and application in swine diets were reported in an earlier study [5]. Apart from that, it also has very high ecological utilization, horticultural, and conservation values. After a long period of exploration and development, *M. lasiocarpa* has been successfully developed as a semi-cultivated plant from a wild plant.

The inflorescence is the most important characteristic and function organ of *M. lasiocarpa*. In addition to being a popular potted plant, landscaping plant, and cut flower, its flower has also been used as an antiphlogistic, hemostatic, anti-enteritis, anti-constipation, detoxification, and anti-gynecological disease agent as well as to alleviate drunkenness [6,7]. Despite the numerous health benefits, there has been little research concerning *M. lasiocarpa* due to its distribution in a small geographical region in China. 

With the development of its artificial cultivation and introduction all over the world, more researchers have begun to focus on its biological and pharmacological activities. In the last two decades, novel compounds have been isolated from the *M. lasiocarpa* flower, which showed significant in vitro anticancer activities and some degree of antimicrobial activity [4,7,8]. These results illustrate the potential of the *M. lasiocarpa* flower as a source of novel drugs, nutraceuticals, and functional foods. However, until now, relatively little was known about its chemical composition and biological activity, which significantly restrained its applications.

Oxidative stress, which is mainly due to the excessive production of reactive oxygen species (ROS), could induce a wide range of chronic disorders, including diabetes, Alzheimer’s disease, cancer, obesity, atherosclerosis, cardiovascular diseases, and inflammatory disorders [9]. Diabetes mellitus is a metabolic disorder caused by hyperglycemia, affecting nearly 10% of the world population, while Alzheimer’s disease (AD) is an irreversible disease caused by severe progressive neurological decline, and its incidence increases with age [10]. Natural antioxidants have exhibited strong functions against oxidative stress, and they possess a wide range of pharmacological activities [11,12]. Hence, natural antioxidants could be an effective preventive agent for oxidative stress-related disorders, such as diabetes and neurodegenerative diseases [13].

To the best of our knowledge, only a few chemical constituents of *M. lasiocarpa* have been reported, and the correlation between these phytochemical components and their biological activities has not been elucidated. Therefore, this is the first study where *M. lasiocarpa* flower was extracted and fractionated by solvents of different polarities, then a comprehensive analysis of the chemical compositions and biological activities of these fractions was performed, including the total phenolic and total flavonoid contents, antioxidant activity, α-glucosidase, acetylcholinesterase, and xanthine oxidase inhibitory abilities, in order to assess the potential in preventing oxidative stress-related diseases. In addition, the phytochemical composition of the most active fraction was determined using HPLC-MS/MS and GC-MS to explore the chemical–activity relationship.

## 2. Results

### 2.1. Total Phenolic and Flavonoid Contents

Table 1 shows that there was a significant difference in the total phenolic content (TPC) and total flavonoid content (TFC) between different fractions. For TPC, this decreased in the order of ethyl acetate fraction (EtOAc), crude methanol extract (CME), aqueous fraction (AF), n-Butanol fraction (n-BuOH), and petroleum ether fraction (PE), and the values were 224.99 ± 6.99, 129.4 ± 2.95, 111.59 ± 2.5, 64.61 ± 1.49, and 12.67 ± 3.15 (mg QE/g DE). TFC decreased in descending order: CME (187.81 ± 9.74 (mg GAE/g DE)), EtOAc (178.95 ± 13.04 mg GAE/g DE), AF (18.85 ± 1.31 mg GAE/g DE), n-BuOH (13.04 ± 0.31 mg GAE/g DE), and PE (0.54 ± 0.03 mg GAE/g). Among the fractions, EtOAc exhibited the highest TPC and TFC values, whereas the TPC and TFC of PE were the lowest, which is consistent with the results reported previously [14].

### 2.2. Antioxidant Activity

#### 2.2.1. DPPH Radical Scavenging Activity

The DPPH· is a stable nitrogen-centered free radical, which serves as the reaction indicator molecule and is widely used in the evaluation of antioxidant activity [15]. The results are shown in Figure 1A, and significant differences between the extract and the fractions produced with different solvents were observed. Among them, EtOAc showed the strongest DPPH radical scavenging activity with IC_50_ values of 22.17 ± 6.01 μg/mL, which had no statistically significant difference when compared with the control Vc and BHT. This was followed by CME, n-BuOH, PE, and AF, with IC_50_ values ranging from 48.21 ± 7.82 to 435.86 ± 17.81 μg/mL.

Correlation analyses (Figure 2) revealed that the significant negative correlation (*r* = −0.67, *p* < 0.01) association was found between the DPPH· scavenging activity with TPC, which was even higher than the correlation with TFC (*r* = −0.57, *p* < 0.05). As DPPH is expressed by IC_50_ value, the lower negative correlation value between DPPH and TPC indicating a stronger DPPH radical scavenging at higher TPC. 

This implies that the DPPH radical scavenging activity of *M. lasiocarpa* may be mainly contributed from phenolics. The results from the principal component analysis (PCA) are shown in Figure 3A, which reveal that the triplicate replicates of different extracts and fractions were well separated. All of the nine indicators were well differentiated by principal coordinates analysis (PCoA) in Figure 3B, and the DPPH· scavenging activity had almost identical principal component coordinates to the α-glucosidase inhibitory activity. Additionally, the other three antioxidant activities had relatively close distances to TPC and TFC.

#### 2.2.2. ABTS Radical Scavenging Activity

The ABTS·^+^ scavenging capacity is based on the ability of hydrogen donating antioxidants to scavenge the long-life radical cation ABTS^+^ by either electron donation or hydrogen electron transfer [16]. As shown in Figure 1B, the EtOAc fraction exhibited the highest scavenging capacity with an IC_50_ value of 12.10 ± 0.48 μg/mL, which was comparable to that of the positive control Vc (10.38 ± 0.57 μg/mL) and significantly stronger than another control, BHT (20.81 ± 0.81 μg/mL). This was followed by CME, where the IC_50_ value was 15.55 ± 1.56 μg/mL, which was also significantly stronger than BHT. This was then followed by AF (46.99 ± 6.86 μg/mL), PE (43.12 ± 0.96 μg/mL), and n-BuOH (35.56 ± 1.12 μg/mL).

It is generally believed that the total phenols, and particularly the total flavonoids, are the main contributors to the scavenging activity estimated by the DPPH and ABTS assays [17], and this was also confirmed by the result of the correlation analysis (Figure 2). A significant correlation (*r* = 0.89, *p* < 0.01) was found between the DPPH and ABTS radical scavenging capacity, which was because the DPPH and ABTS assays had a similar reaction mechanism based on the transfer of a single electron.

#### 2.2.3. Ferric Reducing Antioxidant Power

The ferric reducing antioxidant power (FRAP) assay has demonstrated the ability of antioxidants to reduce iron (III) to iron (II) in a redox-linked colorimetric reaction that involves a single electron transfer [18] and is widely used to screen high-antioxidant ability substances. Based on the standard curve of FeSO_4_ (y = 0.0109x − 0.0968, *R*² = 0.9968), the FRAP results of the samples were calculated and are shown in Figure 1C.

Among all the extract and fractions, EtOAc also exhibited the strongest FRAP (1513.89 ± 90.33 mg FeSO_4_/g DE), which was significantly higher than the other fractions (from 439.49 ± 8.54 to 1513.89 ± 90.33 mg FeSO_4_/g DE). Although lower than the positive control Vc (1741.74 ± 23.19 mg FeSO_4_/g DE), it was nearly double that of another positive control BHT (828.79 ± 81.46 mg FeSO_4_/g DE). From Figure 2, FRAP demonstrated a good and significant correlation at *p* < 0.05 with TPC and TFC, and the correlation coefficients were 0.65 and 0.61, respectively. In addition, the correlations between FRAP and DPPH/ABTS (−0.68, *p* < 0.01) were significantly lower than the DPPH-ABTS correlation (0.89, *p* < 0.01), which was due to differences in their antioxidant mechanism.

#### 2.2.4. Oxygen Radical Absorbance Capacity

The oxygen radical absorbance capacity (ORAC) is a classical in vitro antioxidant assay that measured the radical chain-breaking antioxidant activity, and the mechanism is based on hydrogen atom transfer [19]. According to the Trolox standard curve (y = 1.2372x + 12.774, *R*^2^ = 0.9984), the values of ORAC were obtained and are shown in Figure 1D. EtOAc presented a superior ORAC value of 524.11 ± 30.54 mg TE/g DE, which was even higher than Vc (497.85 ± 18.01 mg TE/g DE). This was followed by PE, n-BuOH, and AF, with the values of 259.37 ± 35.05, 116.84 ± 116.84, and 52.41 ± 3.05 mg TE/g DE, respectively.

A relative association (*r* = 0.57, *p* < 0.05) was seen between ORAC and TFC, which revealed that polyphenols were the main active compounds for ORAC. In contrast, ORAC displayed extremely high negative correlations with the DPPH and ABTS radical scavenging capacity (*r* = −0.79 to −0.84, *p* < 0.01), and FRAP (*r* = 0.88, *p* < 0.01). This is because the ORAC assay represents the sum of the most abundant antioxidants available and is more sensitive, effective, and relevant to human biology when compared with the FRAP, DPPH·, and ABTS·^+^ scavenging assays [16,20].

### 2.3. α-Glucosidase Inhibitory Ability

In clinical applications, α-glucosidase inhibitors are effective in reducing postprandial hyperglycemia by delaying the digestion of carbohydrates and, hence, reducing the absorption of sugars, and they have been recognized as efficient agents in the treatment of type 2 diabetes by virtue of being safe and economical with small side effects [21,22,23].

From Figure 4a, all the extracts and fractions of *M. lasiocarpa* significantly inhibited α-glucosidase in a dose-dependent manner. At the same concentration, EtOAc gave the highest inhibition percentage, followed by CME, AF, PE, and n-BuOH. Their calculated IC_50_ values were also shown in Figure 4b with the values of 18.86 ± 0.44, 79.15 ± 6.03, 143.02 ± 7.66, and 574.86 ± 16.45 μg/mL, respectively. Surprisingly, the α-glucosidase inhibitory activity of EtOAc and CME was significantly stronger than that of the positive control/drug acarbose (125.66 ± 6.13 μg/mL).

This illustrates the therapeutic potential of *M. lasiocarpa* as a natural antidiabetic agent. Correlation analyses revealed that polyphenols were a major contributing factor for the α-glucosidase inhibitory activity of *M. lasiocarpa* (*r* = −0.75, *p* < 0.01). Additionally, a good relationship of α-glucosidase inhibitory activity with both ORAC and FRAP was observed (*r* = −0.81 to −0.91, *p* < 0.01).

### 2.4. Acetylcholinesterase Inhibitory Ability

According to the theory of the “cholinergic hypothesis”, acetylcholinesterase (AChE) inhibitors could increase cholinergic activity by preserving the levels of acetylcholine and further improve the symptoms of AD [24]. Therefore, AChE has become a key target enzyme in AD and has been a hot research field for the treatment of AD in recent years.

As shown in Figure 4c, all the extract and fractions exhibited AChE inhibitory activity in a dose-dependent manner as similar to α-glucosidase inhibitory activity, although they were less active than the positive control/drug galantamine. Among them, AF exerted the strongest inhibitory effect, followed by CME, EtOAc, n-BuOH, and then PE (Figure 4d), and their IC_50_ values were 10.11 ± 1.38, 14.40 ± 1.10, 31.63 ± 11.06, 110.01 ± 11.23, and 149.56 ± 20.08 μg/mL, respectively. In agreement with the AF fraction, the CME of *M. lasiocarpa* also exhibited high AChE inhibitory activity. Correlation analyses showed a poor relationship of AChE inhibitory activity with other biological activities.

### 2.5. Xanthine Oxidase Inhibitory Ability

Xanthine oxidase (XO), a key enzyme in the purine metabolism, can catalyze the oxidation of hypoxanthine to xanthine, then to uric acid, eventually leading to hyperuricemia and gout [25]. According to the Figure 4e, all the extract and fractions demonstrated substantial XO inhibitory activity compared to allopurinol as a positive control (*p* < 0.05). The IC_50_ values are shown in Figure 4f in the order of CME (5.23 ± 0.35) > PE (7.43 ± 3.77 μg/mL) > EtOAc (7.50 ± 0.16 μg/mL) > n-BuOH (12.64 ± 1.12 μg/mL) > AF (14.32 ± 2.08 μg/mL) > allopurinol (24.85 ± 0.46 μg/mL).

From Figure 2, the XO inhibitory activity was negatively correlated with TFC (*r* = −0.66, *p* < 0.01). This is because flavonoids have been reported to possess the ability to act as active inhibitors of xanthine oxidase by competitively hindering the enzyme actions [26]. Beyond this, significant negative correlations between the XO inhibitory activity with FRAP and ORAC were observed (*r* = −0.61 to −0.64, *p* < 0.05).

### 2.6. HPLC-MS/MS Analysis

Based on the results of the antioxidant properties and enzyme inhibitory activities, the EtOAc fraction with the higher content of bioactive component and biological activity was selected for the HPLC-MS/MS analysis. The total ion chromatogram and the molecular structure of the compounds identified are depicted in Figure 5, and all information needed for the compound assignment is summarized in Table 2. A total of ten compounds were separated, and one phenolic acid, one terpenoid, and seven flavonoids were identified, including catechin (1), daphuribirin D (2), epicatechin derivative (3), an unknown compound (4), vaterioside A (5 or 6), 1,6,2′,6′-tetraacetyl-3-*O*-p-coumaroylsucrose (7), 1,2′,3′,4′,6′-pentaacetyl-3-*O*-p-coumaroylsucrose (8), 6-methoxy-1,2′,3′,4′,6′-pentaacetyl-3-*O*-p-3,4,5-trihydroxy cinnamoylsucrose (9), and 1,6,2′,3′,4′,6′-hexaacetyl-3-*O*-p-coumaroylsucrose (10).

Among them, compounds 5–10 were first identified in *M. lasiocarpa*, and their structural formulas are shown in Figure 5B. Many of the compounds presented certain biological activities. For example, research reported the α-glucosidase inhibitory activity and radical scavenging activity of catechin [27]. Daphuribirin D belongs to bifuranocoumarins, which has demonstrated wide-ranging bioactivities, inclusive of analgesic, anticoagulant anti-HIV, anti-inflammatory, antimicrobial, antineoplastic, antioxidant, and immunomodulatory effects, due to its structural diversity [28].

Moreover, the epiatechin derivative is known as a bioactive molecule with anticancer, anti-oxidant, hepatoprotective, anti-inflammatory, and anti-microbial properties [29]. Overall, the compounds identified were mainly flavonoids, which may be responsible for the excellent antioxidant activities and α-glucosidase inhibitory activity of the EtOAc fraction.

**Table 2 molecules-26-04472-t002:** Identified or tentatively identified compounds of the EtOAc fraction of *M. lasiocarpa* by HPLC-MS/MS.

	RT (time)	MS [M-H]	Molecular Formula	Molecular Weight	Concentration (ppm)	MS/MS Fragments	Name of Compounds	Classification	Reference
1	7.673	289.0682	C_15_H_14_O_6_	290.079	12.28	203.0789	Catechin	Phenolic acids	[30]
2	8.056	571.1620	C_32_H_28_O_10_	572.1682	−1.8	117.03,145.0261, 252.0938	Daphuribirin D	Terpenoids	[31]
3	8.129	723.4958	C_37_H_72_O_13_	724.4973	−7.98	207.1456,660.0386, 677.4916	Epiatechin derivative	Flavonoids	[32]
4	8.229	836.5792	C_51_H_81_O_9_	837.5822	1.89	109.2067,230.8014, 790.5704	unknown	-	-
5	8.579	613.1719	C_34_H_30_O_11_	614.1788	−0.59	117.0314, 145.0279, 146.0321	Vaterioside A	Flavonoids	[33]
6	8.746	613.1719	C_34_H_30_O_11_	614.1788	−0.59	145.0279, 146.0321, 163.038	Vaterioside A isomer	Flavonoids	[33]
7	9.864	655.1833	C_29_H_36_O_17_	656.1952	7.12	117.0334, 145.0282, 163.0403	1,6,2′,6′-Tetraacetyl-3-*O*-p-coumaroylsucrose	Flavonoids	[34]
8	10.948	697.1947	C_31_H_38_O_18_	698.1985	5.5	145.0283, 163.038, 655.1838	1,2′,3′,4′,6′-Pentaacetyl-3-*O*-p-coumaroylsucrose	Flavonoids	[35]
9	11.131	743.1991	C_32_H_40_O_20_	744.2113	6.61	145.028, 163.0393	6-Methoxy-1,2′,3′,4′,6′-pentaacetyl-3-*O*-p-3,4,5-trihydroxy cinnamoylsucrose	Flavonoids	[36]
10	12.449	739.2032	C_33_H_40_O_9_	740.2164	7.97	117.0357, 145.0281, 146.03	1,6,2′,3′,4′,6′-Hexaacetyl-3-*O*-p-coumaroylsucrose	Flavonoids	[37]

### 2.7. GC-MS Analysis

Chemical analysis of the EtOAc fraction was further performed by GC-MS. The total ion chromatogram is shown in Figure 6. According to the corresponding mass spectral databases, 15 major compounds were identified and listed in Table 3, including seven acids, one alcohol, and seven esters. The relative content was in the following order: 9-octadecenoic acid (Z)-, methyl ester (29.96%), trans-13-octadecenoic acid (8.86%), 9-octadecenoic acid, methyl ester, (E)- (8.45%), cis-13-eicosenoic acid, methyl ester (8.43%), hexadecanoic acid, methyl ester (5.16%), methyl stearate (4.69%), eicosanoic acid ME P891 (3.65%), hexacosyl pentafluoropropionate (3.47%), phthalic acid, butyl hept-3-yl ester (1.88%), cis-13-octadecenoic acid (1.47%), methyl 9-cis,11-transoctadecadienoate (1.06%), trans-á-santalol (1.04%), 1,4-benzenedicarboxylic acid, dimethyl ester (0.95%), n-hexadecanoic acid (0.81%), cis-11-eicosenoic acid, and methyl ester (0.78%). 

These compounds, therefore, also might be the primary reason for the biological activities of *M. lasiocarpa*. Interestingly, the 9-octadecenoic acid (Z)-, methyl ester, was dominant in the entire GC-MS chromatogram, which was found to have antimicrobial activity, for instance, against *S. aureus*, *E. coli* and *M. smegmatis* [38]. It can be clearly speculated that fatty acid of *M. lasiocarpa* might be responsible for antifungal activity and contribute to the prevention of infectious diseases.

## 3. Discussion

Despite being a flower with high cultural, edible, medicinal, and ornamental values, the chemical components and bioactivities of *M. lasiocarpa* have been scarcely evaluated. In this study, the crude methanol extract of *M. lasiocarpa* and its fractions by solvents of different polarities were prepared. The TPC, TFC, and antioxidant activities, including the DPPH and ABTS radical scavenging capacity, FRAP, and ORAC of the extract and fractions were determined. The therapeutic potential in the treatment of diabetes, Alzheimer’s disease, and hyperuricemia were assessed by testing its inhibitory effect on α-glucosidase, acetylcholinesterase, and xanthine oxidase. Finally, the chemical compositions of the fraction with the best biological activity were investigated using both HPLC-MS/MS and GC-MS techniques.

Among all the fractions, EtOAc possessed the highest bioactive component content and better biological activity. In addition to the highest TPC and higher TFC, its IC_50_ values of the DPPH and ABTS radical scavenging abilities, FRAP values, and ORAC values were significantly higher than the other fractions, which were comparable if not better than that of the positive control Vc. Furthermore, each other fraction had different degrees of antioxidant activity. As reported in several works of literature, the antioxidant activity of the plant extracts is correlated with their total phenolic content and total flavonoid content, and the solvents system has great effects on the phenolic, flavonoid contents, and antioxidant activities [39,40,41].

For the α-glucosidase, acetylcholinesterase, and xanthine oxidase inhibitory abilities, the EtOAc, AF, and CME fractions displayed the strongest activity, respectively. Previous studies showed that a few edible or medicinal plants, including hawthorn fruit [42], *Lycopodiastrum casuarinoides* [43], and *Corchorus depressus* [44], were reported with α-glucosidase and AChE inhibitory activities, in agreement with *M. lasiocarpa*. In combination, these three fractions exhibited relatively strong activity in these three enzyme inhibition experiments, which may be related to their higher phenolic or flavonoid contents [45].

By using HPLC-MS/MS and GC-MS approaches, a total of 9 and 15 compounds were identified in the EtOAc fraction, respectively. Compared to the previous studies [4,7], more new compounds in *M. lasiocarpa* were identified. Most of them were flavonoids, which also explains the higher α-glucosidase and acetylcholinesterase inhibitory abilities of the EtOAc fraction as the results showed a highly significant correlation of TFC with the α-glucosidase, acetylcholinesterase, and xanthine oxidase inhibitory abilities.

In summary, *M. lasiocarpa* exhibited excellent antioxidant activity, α-glucosidase, acetylcholinesterase, and xanthine oxidase inhibition activities, especially for its EtOAc fraction, which may be related to its high contents of phenolic and flavonoid compounds. The results suggest that *M. lasiocarpa* has the potential to be exploited as a natural source of preventive agent for diabetes mellitus, Alzheimer’s disease, and gout [21,22,23,24,25].

However, considering that these data were obtained from in vitro assays, these extract and fractions without digestion treatment would be unlikely to maintain the chemicals and activities from ingestion until arriving at the target organ in addition to the isolation and purification of the actual active compounds. Additional in vivo experiments are needed to determine and confirm the biological uses before proceeding to human intervention.

## 4. Materials and Methods

### 4.1. Standards and Reagents

The 2,2′-azino-bis-(3-ethylbenzothiazoline-6-sulfonic acid) (ABTS), 2,2-diphenyl-1-picrylhydrazyl (DPPH), 2,4,6-Tris(2-pyridyl)-s-triazine (TPTZ), 2,2′-azobis(2-amidinopropane) dihydrochloride (AAPH), fluorescein sodium, acetylthiocholine iodide**,** vitamin C (Vc), and 2,6-di-tert-butyl-4-methylphenol (BHT) were obtained from Aladdin Biotechnology (analytical grade, China). Other analytical grade chemicals, including methanol, petroleum ether, ethyl acetate, and n-butanol were purchased from Sinopharm Chemical Reagent Co., Ltd. (Shanghai, China). 

Chromatographic acetonitrile was purchased from Merck (Darmstadt, Germany). α-glucosidase (from saccharomyces cerevisiae), xanthine oxidase (XO), acetylcholinesterase (AChE), acarbose, galantamine (GALM), allopurinol, 4-methylumbelliferyl-β-D glucuronide (4-MUG), 5,5′-dithiobis (2-nitrobenzoic acid) (DTNB), and acetylthiocholine iodide (ATCI) were purchased from Sigma-Aldrich (St. Louis, MO, USA).

### 4.2. Sample Preparation

Plants of *M. lasiocarpa* were collected in Southwest Forestry University, Kunming, Yunnan Province, China, in August 2016. After collection, the flowers of *M. lasiocarpa* were immediately transferred into a cool box with ice blocks before transportation to the laboratory. The samples were stored at −80 °C and then freeze-dried and smashed into powder. The powdered sample (100 g) was mixed with 2 L of 70% methanol, followed by an ultrasonic extraction for 1 h at 50 °C with an ultrasonic power of 300 W. After centrifugation at 4000 rpm for 20 min, the filtration residue was re-extracted again. 

The supernatants were combined and evaporated under a vacuum at 50 °C to yield the crude methanol extract (CME). Then, the CME was dissolved in 500 mL H_2_O and fractionated via liquid–liquid partitioning with solvents of different polarities successively at a 1:1 volume ratio, finally producing the petroleum ether (PE) fraction, ethyl acetate (EtOAc) fraction, n-butanol (n-BuOH) fraction, and aqueous fraction (AF). All the extract and fractions were stored at −18 °C until further use.

### 4.3. Total Phenolic and Flavonoid Contents

The total phenolic content (TPC) was determined by the Folin–Ciocalteu method [46]. Briefly, a 40 µL sample at appropriate concentration and 25 µL of Folin–Ciocalteu reagent were mixed in a 96-well microplate, and then 200 µL of 7.5% Na_2_CO_3_ solution (*w*/*v*) was added and kept in the dark at room temperature for 25 min. The absorbance at 765 nm was determined, and the TPC was counted from a calibration curve plotted against gallic acid (20–100 μg/mL). The results were expressed as mg of gallic acid equivalents per g of dry extract (mg GAE/g DE).

The total flavonoid content (TFC) was measured using the AlCl_3_ colorimetric method [47]. A 100 µL suitable concentration of sample solution was pipetted onto a 96-well plate, then 100 µL AlCl_3_·6H_2_O solution (2%) was added, and the absorbance was measured at 430 nm after 6 min. Quercetin (2.0–12.0 μg/mL) was used as the standard, and the results are expressed as mg of quercetin equivalents per g of dry extract (mg QE/g DE).

### 4.4. Antioxidant Activity Determination

#### 4.4.1. DPPH Radical Scavenging Activity

The DPPH· scavenging activity was assessed by referring to the reported method [48]. A 100 µL properly diluted sample solution was mixed with 100 µL of DPPH solution (0.15 mmol/L) on a 96-well plate, after storing at room temperature in dim light for 30 min, and then the absorbance value (*As*) was read at 517 nm. Methanol was used to replace the sample as a negative control (*Ab*), while Vc and BHT were the positive controls. The percentage of DPPH radical clearance was calculated using the following Formula (1), and the DPPH· scavenging activity was expressed as the IC_50_ value (μg/mL), which was obtained by plotting the percentage of scavenging versus the concentration.
(1)DPPH radical scavenging rate (%)=[(Ab−As)/Ab]×100%

#### 4.4.2. ABTS Radical Scavenging Activity

The ABTS·^+^ scavenging activity assay was performed as described by Arts et al. [49]. The ABTS solution (7 mM) was mixed with potassium sulfate (2.45 mM) and incubated for 12 h at room temperature in the dark. Before the assay, the ABTS working solution was diluted with methanol to the absorbance of 0.70 ± 0.02 at 734 nm. A 50-µL appropriate concentration of sample solution was mixed with 200 µL of ABTS working solution on a 96-well plate and kept for 6 min at room temperature, and then the absorbance at 734 nm was measured. Methanol was used instead of the sample as a negative control (*Ab*), while Vc and BHT were used as positive controls. The scavenging rates were calculated according to Equation (1), and the results are expressed as IC_50_ values (μg/mL).

#### 4.4.3. Ferric Reducing Antioxidant Power Assay

The ferric reducing antioxidant power (FRAP) method was carried out according to the method of [50]. TPTZ (10 mM), FeCl_2_ (20 mM), and acetate buffer (300 mM) were mixed in the ratio of 1:1:10 to prepare a fresh working FRAP reagent. A 20-µL sample solution with a suitable concentration and 300 µL FRAP working solution were loaded onto a 96-well plate, and the absorbance was read at 593 nm after incubation for 10 min at 37 °C. Methanol was used as a negative control, and Vc and BHT (40 μg/mL) were used as positive controls. Ferrous sulfate (20–100 μg/mL) was taken as a standard for the preparation of the standard curve, and the FRAP value was expressed as mg of FeSO_4_ per gram of dry extract (mg FeSO_4_/g DE).

#### 4.4.4. Oxygen Radical Absorbance Capacity (ORAC) Assay

The ORAC was performed as recommended by a previous report of Huang [51]. A 25-µL properly diluted sample was mixed with 200 µL of fluorescein on a black 96-well fluorescence microplate at 37 °C for 10 min, and then 50 µL of AAPH was added into each well. The fluorescence generated was read at 535 nm emission and 485 nm excitation every 1.5 min for 2 h using a fluorescence microplate reader. Trolox (10–100 μg/mL) was used as the standard. Methanol was used as a negative control, and Vc and BHT were used as positive controls. The results of ORAC values were calculated using the area under the curve (AUC) with the Trolox standard curve and are expressed as mg of Trolox equivalents (TE)/g dry extract (mg TE/g DE).

### 4.5. Enzyme Inhibitory Ability

#### 4.5.1. Inhibition of α-Glucosidase

The α-glucosidase inhibitory activity was evaluated following a previous method [52]. A 50-µL suitably diluted sample solution was mixed with 50 µL of α-glucosidase (0.1 U/mL, pH = 6.9) in a black microtiter 96-well plate and incubated for 10 min at 25 °C. Then, 50 μL of 4-MUG (0.84 mM) was added and incubated for 25 min at 25 °C. 

The reaction was then terminated by adding a 100 μL glycine-NaOH buffer (100 mM, pH 10.6). After shaking on an orbital shaker for 30 s, the fluorescence was measured at *λ*ex 355 nm and *λ*ex 460 nm. Methanol was used as a negative control, and acarbose was used as a positive control. The α-glucosidase inhibitory ability was expressed as the IC_50_ value (µg/mL). The α-glucosidase inhibition rate was counted as follows:(2)$$\mathrm{Inhibition}\;\mathrm{rate}\;\left(\%\right)=\frac{Ab\;-\;(As-Ac)}{\;Ab}\times 100\%\;$$
where *Ab*, *As*, and *Ac*, respectively, represent the fluorescence values of the reagent blank (without sample), test samples (with all reagents), and negative control (without α-glucosidase).

#### 4.5.2. Inhibition of Acetylcholinesterase

The acetylcholinesterase (AChE) inhibitory activity assay was measured by the Ellman colorimetric method [53]. A 50-µL appropriate diluted sample solution, 20 μL of acetylcholine (0.2 U/mL), and 90 µL of Ellman’s reagent (containing 15 µL of 15 mM ATCI and 75 µL of 3 mM DTNB) were added to a 96-well plate in darkness for 10 min. Finally, 20 μL of 0.2 U/mL AChE was added and kept in darkness for 5 min, and the absorbance was read at 405 nm (*As*). Methanol in place of the sample was used as a blank (*Ab*), phosphate buffer in place of acetylcholine was used as a negative control (*Ac*), and galanthamine was used as a positive control. The inhibition rate calculated was similar to Equation (2).

#### 4.5.3. Inhibition of Xanthine Oxidase

The method for measuring the xanthine oxidase (XO) inhibitory activity was slightly modified from a previous method [54]. A 50-µL sample, 60 μL phosphate buffer (70 mM, PH = 7.5), and 50 μL of XO were mixed for 15 min at 25 °C, and the reaction was started by the addition of 60 μL xanthine solution (1.5 mM). After incubation for 8 min at 25 °C, the absorbance was read at 295 nm (*As*). Methanol was used as the reagent blank (*Ab*), and phosphate buffer was used as a negative control (*Ac*). Allopurinol was used as a positive control. The XO inhibitory ability was expressed as the IC_50_ value (µg/mL).

### 4.6. HPLC-MS/MS Analysis

The HPLC-MS/MS analysis was performed according to the previously described method [55]. The equipment consisted of an Agilent1290 UPLC coupled to a diode array detector, a triple quadruple-ion trap mass analyzer, an electrospray ionization (ESI) source, and an Agilent Technologies 6538 OHD Accurate Mass MS/MS system (Agilent Technologies, CA, USA). The separation was performed by Agilent ZORBAX SB-C18 (4.6 × 250 mm, 5 μm) column, with mobile phase A (deionized water + 0.1% formic acid) and mobile phase B (acetonitrile). To obtain the best separation in a short time, the gradient program and flow rate were optimized. 

The optimum linear-gradient started with 10% B and the conditions of the mobile phase B were: 0–10 min, 15% B; 10–20 min, 20% B; 20–30 min, 27% B; 30–40 min, 40% B, and finally the column was washed with 100% B for 5 min. After filtering through a 0.45 µm membrane, 20 µL of the sample was injected into the system. UV spectra from 200 to 400 nm were recorded for peak characterization. The automatic MS/MS experiment was fragmented by adjusting the collision energy as follows: *m*/*z* < 200, 10, and 20 eV; *m*/*z* 200–400, 20, and 30 eV; *m*/*z* 400–600, 30, and 40 eV; *m*/*z* > 600, 40, and 50 eV; and *m*/*z* > 700, 40, 50, and 60 eV. 

The MS data were processed with MassHunter software (Agilent Technologies, Santa Clara, CA, USA) and searched using the Generate Molecular Formula^TM^ editor to obtain the possible molecular formula of each precursor and product. The compound at each peak was identified according to its precursor ion, molecular weight, fragmentation pattern, and retention time, and we matched these data with that reported in available references.

### 4.7. GC-MS Analysis

The fraction with the best biological activity was analyzed with gas chromatography coupled to a mass spectrometer (GC-MS, Trace1300/ISQ, Thermo Fisher, Waltham, MA, USA) [56]. A capillary HP-5 column (30.00 m × 0.25 mm × 0.25 µm) was used for sample separation. The injector temperature and injection volume were 250 °C and 1 μL, respectively. The initial temperature was 50 °C for 2 min, followed by an increase of 3 °C/min up to 160 °C, after which it increased to 220 °C with 5 °C/min. Helium was used as carrier gas with a flow rate of 1 mL/min, and the ionization energy was 70 eV.

### 4.8. Statistical Analysis

All data are presented as the mean ± standard deviation (SD) with triplicate analyses of the same sample. Correlation analyses were performed using SPSS Version 22.0 by multiple linear regression. *p* < 0.05 was considered as significant (*), and *p* < 0.01 was highly significant (**). The IC_50_ values were calculated with the polynomial fit using Origin 2018 software.

## Figures and Tables

**Figure 1 molecules-26-04472-f001:**
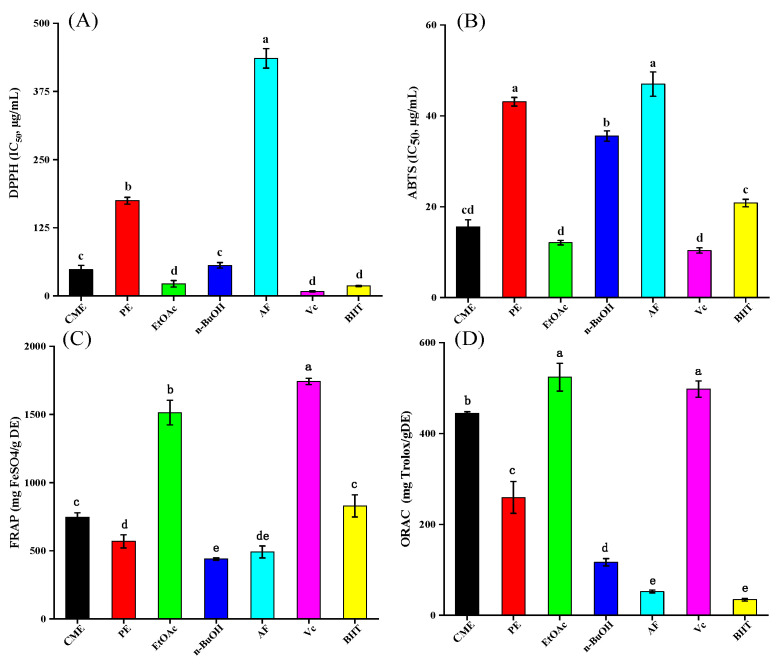
The antioxidant activity of samples or controls. (**A**) DPPH· scavenging activity (DPPH). (**B**) ABTS·^+^ scavenging ability (ABTS). (**C**) Ferric reducing antioxidant power (FRAP). (**D**) Oxygen radical absorbance capacity (ORAC). CME = crude methanol extract; PE = petroleum ether fraction; EtOAc = ethyl acetate fraction; n-BuOH= n-Butanol fraction; and AF =  aqueous fraction. The columns with different lowercase letters represent a significant difference at *p* < 0.05.

**Figure 2 molecules-26-04472-f002:**
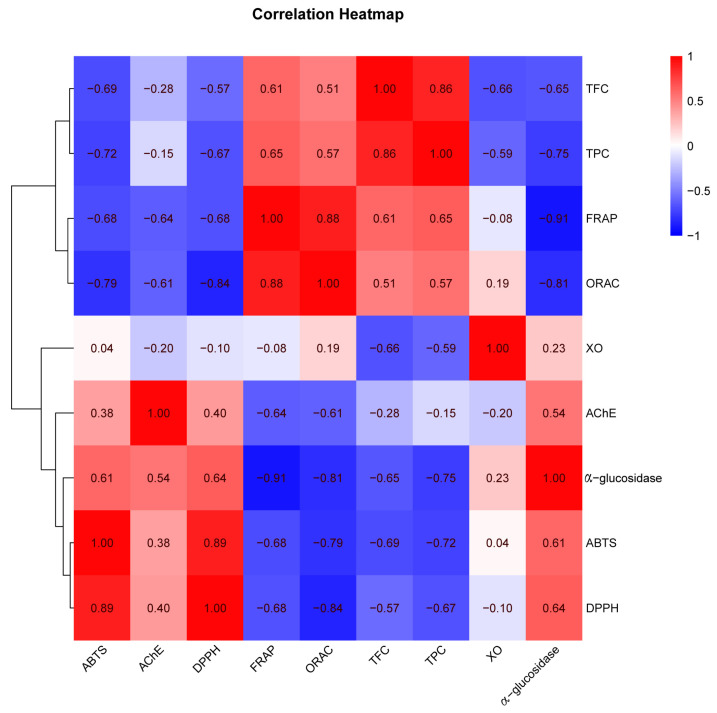
Heatmap graph of correlation analysis. *p*-value (ranging from −1 to 1) and corresponding color (red to blue) represent the magnitude of the Pearson correlation coefficient (*r*) in the heatmap; TPC = total phenolic content; TFC = total flavonoid content; DPPH = DPPH· scavenging activity; ABTS = ABTS·^+^ scavenging ability; FRAP = ferric reducing antioxidant power; ORAC = oxygen radical absorbance capacity; α-glucosidase = the IC_50_ value for α-glucosidase inhibitory activity; AChE = the IC_50_ value for acetylcholinesterase inhibitory activity; and XO = the IC_50_ value for xanthine oxidase inhibitory activity. The different letters represent a significant difference, *p* < 0.05.

**Figure 3 molecules-26-04472-f003:**
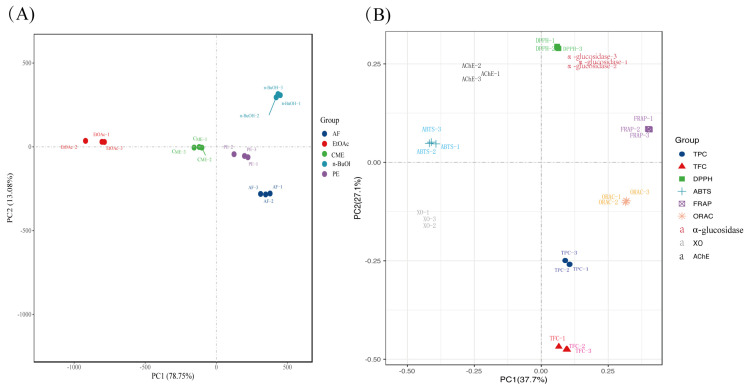
Principal component analysis (PCA) of different samples and indicators. (**A**) PCA of five *M. lasiocarpa* extract and fractions. (**B**) principal co-ordinates analysis (PCoA) of nine indicators of active ingredient content and biological activity. CME = crude methanol extract; PE =petroleum ether fraction; EtOAc = ethyl acetate fraction; n-BuOH= n-Butanol fraction; AF  =  aqueous fraction; TPC = total phenolic content; TFC = total flavonoid content; DPPH = DPPH· scavenging activity; ABTS = ABTS·^+^ scavenging ability; FRAP = ferric reducing antioxidant power; ORAC = oxygen radical absorbance capacity; α-Glucosidase = the IC_50_ value for α-glucosidase inhibitory activity; AChE = the IC_50_ value for acetylcholinesterase inhibitory activity; and XO = the IC_50_ value for xanthine oxidase inhibitory activity.

**Figure 4 molecules-26-04472-f004:**
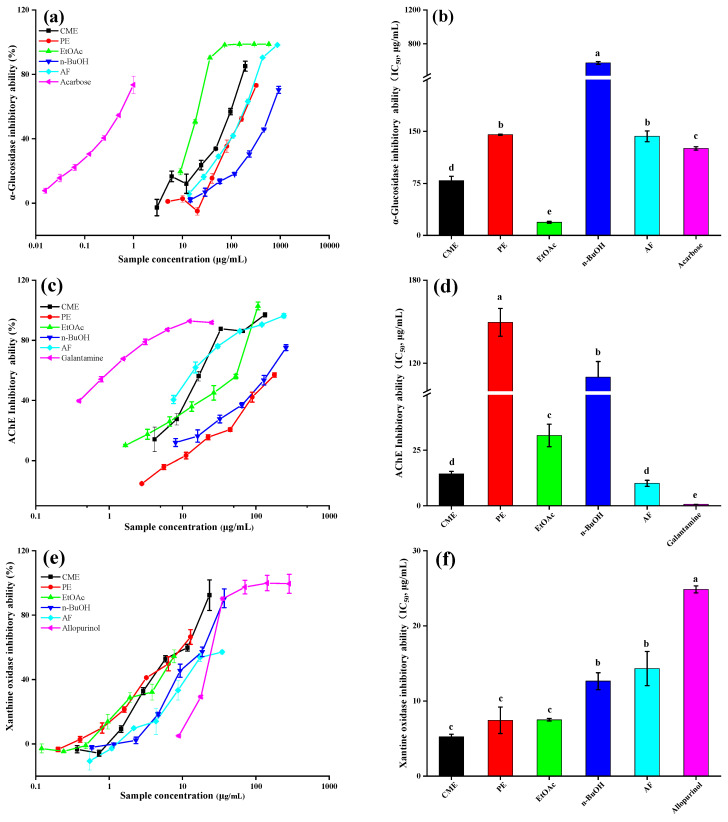
The enzyme inhibitory ability of the *M. lasiocarpa*. (**a**) The inhibition rate of the α-glucosidase with different concentrations. (**b**) The IC_50_ values of different samples toward α-glucosidase inhibitory activity. (**c**) The inhibition rate of the XO with different concentrations. (**d**) The IC_50_ values of different samples toward XO inhibitory activity. (**e**) The inhibition rate of the AChE with different concentrations. (**f**) The IC_50_ values of different samples toward AChE inhibitory activity. CME = crude methanol extract; PE =petroleum ether fraction; EtOAc = ethyl acetate fraction; n-BuOH= n-Butanol fraction; and AF  =  aqueous fraction. IC_50_ = half maximal inhibitory concentration. Columns with different letters indicate a significant difference (*p* < 0.05).

**Figure 5 molecules-26-04472-f005:**
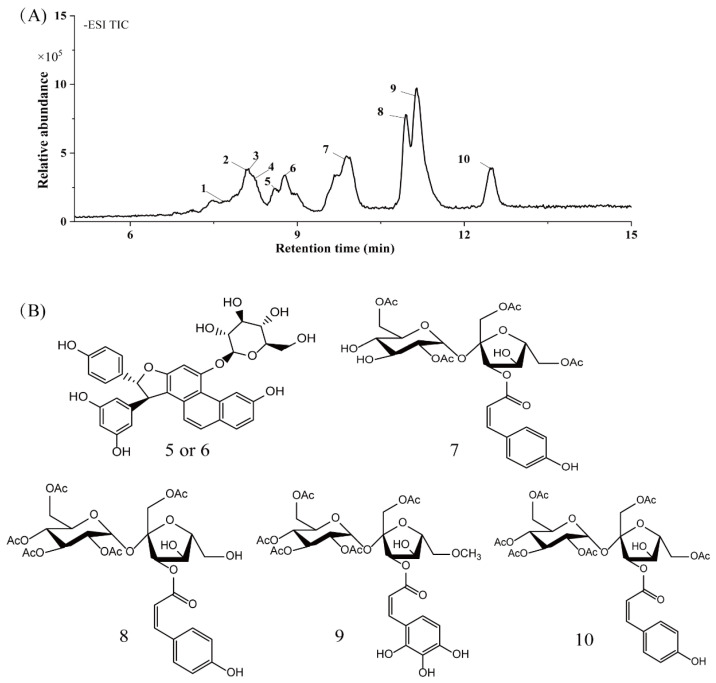
(**A**) The HPLC-MS/MS total ion current (TIC) chromatogram. The numbers 1–10 above the peaks indicate the location of the identified compounds. (**B**) Chemical structures of the 5–10 compounds. 5 or 6: vaterioside A; 7: 1,6,2′,6′-tetraacetyl-3-*O*-p-coumaroylsucrose; 8: 1,2′,3′,4′,6′-pentaacetyl-3-*O*-p-coumaroylsucrose; 9: 6-methoxy-1,2′,3′,4′,6′-pentaacetyl-3-*O*-p-3,4,5-trihydroxy cinnamoyl sucrose; and 10: 1,6,2′,3′,4′,6′-hexaacetyl-3-*O*-p-coumaroylsucrose.

**Figure 6 molecules-26-04472-f006:**
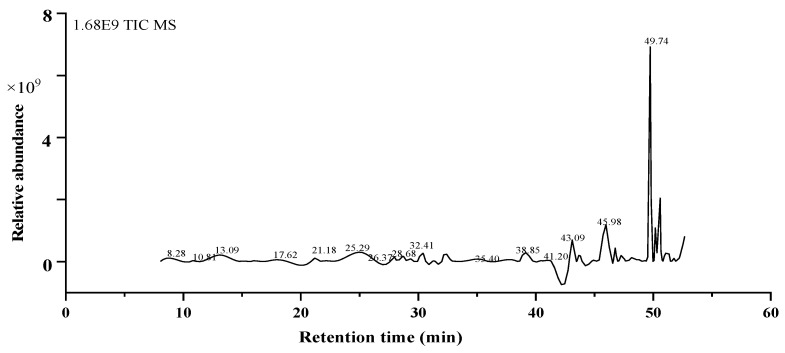
The GC-MS total ion current (TIC) chromatogram. The numbers above the peaks indicate the retention time for each compound.

**Table 1 molecules-26-04472-t001:** The TPC and TFC values of *M. lasiocarp* extracts.

Samples	TPC	TFC
CME	129.40 ± 2.95 ^b^	187.81 ± 9.74 ^a^
PE	12.67 ± 3.15 ^e^	0.54 ± 0.03 ^c^
EtOAc	224.99 ± 6.99 ^a^	178.95 ± 13.04 ^a^
n-BuOH	64.61 ± 1.49 ^d^	13.04 ± 0.31 ^bc^
AF	111.59 ± 2.50 ^c^	18.85 ± 1.31 ^b^

TPC = total phenolic content (mg GAE/g DE); TFC = total flavonoid content (mg QE/g DE); CME = crude methanol extract; PE = petroleum ether fraction; EtOAc = ethyl acetate fraction; n-BuOH = n-Butanol fraction; and AF =  aqueous fraction. The means with different lowercase letters in the same column are significantly different (*p* < 0.05).

**Table 3 molecules-26-04472-t003:** The results of GC–MS analysis of the EtOAc fraction.

	RT(time)	Name of Compounds	Molecular Formula	Area	Area(%)	Structure
1	30.44	Trans-á-santalol	C_15_H_24_O	240,544,370.50	1.04	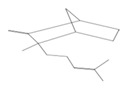
2	32.17	1,4-Benzenedicarboxylic acid, dimethyl ester	C_10_H_10_O_4_	218,812,929.34	0.95	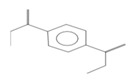
3	42.89	Cis-13-eicosenoic acid, methyl ester	C_21_H_40_O_2_	1405,375,753.9	8.43	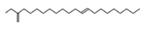
4	43.78	Cis-11-eicosenoic acid, methyl ester	C_21_H_40_O_2_	181,015,792.24	0.78	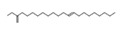
5	45.72	Eicosanoic acid ME P891	C_21_H_42_O_2_	842,605,575.83	3.65	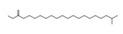
6	45.97	Hexadecanoic acid, methyl ester	C_17_H_34_O_2_	1191,145,281.60	5.16	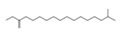
7	46.76	Phthalic acid, butyl hept-3-yl ester	C_19_H_28_O_4_	434,470,530.88	1.88	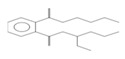
8	46.86	n-Hexadecanoic acid	C_16_H_32_O_2_	186,578,734.29	0.81	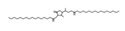
9	49.73	9-Octadecenoic acid (Z)-, methyl ester	C_19_H_36_O_2_	6915,810,529.98	29.96	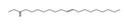
10	49.82	9-Octadecenoic acid, methyl ester, (E)-	C_19_H_36_O_2_	1950,734,475.62	8.45	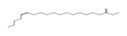
11	50.17	Methyl stearate	C_19_H_38_O_2_	1083,118,959.91	4.69	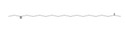
12	50.58	trans-13-Octadecenoic acid	C_18_H_34_O_2_	2044,402,879.86	8.86	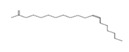
13	50.65	cis-13-Octadecenoic acid	C_18_H_34_O_2_	338,345,248.07	1.47	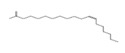
14	51.35	Methyl 9-cis,11-transoctadecadienoate	C_19_H_34_O_2_	243,690,268.64	1.06	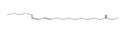
15	52.66	Hexacosyl pentafluoropropionate	C_29_H_53_F_5_O_2_	802,126,926.87	3.47	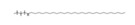

## Data Availability

The remaining data are available on request from the corresponding author.

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
