# Peer review of "Phytochemical Composition, Antioxidant Activity, and Enzyme Inhibitory Activities (α-Glucosidase, Xanthine Oxidase, and Acetylcholinesterase) of Musella lasiocarpa"

_molecules, 2021, doi:10.3390/molecules26154472_

Round 1

Reviewer 1 Report

Phytochemical composition, antioxidant activity, and enzymes inhibitory activities (α-glucosidase, xanthine oxidase, and acetylcholinesterase) of Musella lasiocarpa 4

This is a standard article; just another plant was used for extraction and determination of bioactivity.

Abstract:  According to the abstract, “Taken together, the results indicated that Musella lasiocarpa could be exploited as a natural source with excellent antioxidants in the prevention of type 2 diabetes, Alzheimer’s disease, hyperuricemia, and their complications.”

This is a very strong conclusion that the plant can be used for all so serious diseases.

Very important to underline what is new in your article in comparison with the published once just very shortly.

Introduction: Some information is not so important, because it is known about the antioxidants, etc for a long time, just underline as you put the last sentence.

“This is the first study on this aspect, and it will provide a scientific basis for further utilization of M. lasiocarpa in the food and nutraceutical industry.

This is a common conclusion, but what is new in this report and why was undertaken this research.

Results: There are many abbreviations which were used for the first time and not disclosed.  Please, check.

The data of antioxidant assays   are presented in different units, then how it is possible to compare, just recalculate or use the standard curve of Trolox equivalents.

Fig. 1. In the legend not abbreviations are not shown, just some

Fig. 2. Correct the ABTS symbols

Fig. 3 is not so sharp and difficult to read.

Discussion: the last sentence is a same as in the abstract. Based only on the antioxidant and inhibitory tests and obtained data and not using any in vitro experiments is difficult to do such conclusion. Put as well some references in order to make stronger your general conclusion.

M&M: How the flowers were prevented from oxidation before freeze drying

 As I mentioned above the antioxidant assays were done using different units, therefore it is difficult to compare different results.

Check carefully the text.

Author Response

Dear Editor in Chief Prof. Freda Sun and Reviewers:

Thank you for giving us the opportunity to revise our manuscript and your valuable comments concerning our manuscript entitled “Phytochemical composition, antioxidant activity, and enzymes inhibitory activities (α-glucosidase, xanthine oxidase, and acetylcholinesterase) of Musella lasiocarpa” (ID: molecules-1292480). The comments are very helpful for improving our manuscript. We have revised the manuscript carefully according to reviewers’ comments point by point. All modifications to the manuscript are highlighted in red for your further comments. The responses to the reviewer’s comments are as follows:

Reviewer 1:

  1. Abstract: According to the abstract, “Taken together, the results indicated that Musella lasiocarpa could be exploited as a natural source with excellent antioxidants in the prevention of type 2 diabetes, Alzheimer’s disease, hyperuricemia, and their complications.” This is a very strong conclusion that the plant can be used for all so serious diseases. Very important to underline what is new in your article in comparison with the published once just very shortly.

Response: Thanks for your kind comment. We fully agree with the reviewer and we have revised the conclusion in the “Abstract”. Please see lines 34-35.

  1. Introduction: Some information is not so important, because it is known about the antioxidants, etc for a long time,

Response: As advised, we have “streamlined” the introduction of antioxidants. Please see lines 69-70.

  1. Introduction: “This is the first study on this aspect, and it will provide a scientific basis for further utilization of M. lasiocarpa in the food and nutraceutical industry”. This is a common conclusion, but what is new in this report and why was undertaken this research.

Response: Thank you for your advice. This part has been revised to clarify why we conducted this study and the innovation of our study compared with other reports. Please look at the last paragraph of the article in lines 73-82.

  1. Results: There are many abbreviations which were used for the first time and not disclosed.  Please, check.

Response: Thank you for your valuable comments. Because many abbreviations in the results are listed in the “4. Materials and Methods” section, we ignore the reading order, which is our negligence. After checking, the unmarked abbreviations have been added to "2. Results". Please see lines 85-88.

  1. Results: The data of antioxidant assays are presented in different units, then how it is possible to compare, just recalculate or use the standard curve of Trolox equivalents.

Response: Thank you for the comment. We refer to different antioxidant methods to calculate and evaluate the antioxidant capacity, and all of them are described in the methods section. The DPPH and ABTS are expressed by IC50 value. Please see lines 385 and 397, respectively. Ferric reducing antioxidant power assay and oxygen radical absorbance capacity assay are expressed by FeSO4 and Trolox equivalents, please see lines 406 and 416, respectively. The comparison was done among the extract and fractions based on data collected using the same antioxidant method to identify the fraction with the highest antioxidant activities.  

  1. Results: Fig. 1. In the legend not abbreviations are not shown, just some

Response: Thank you for your valuable comments. Indeed, as you said, the uniform legend abbreviation is more conducive to the consistency of the picture. Figure 1 (A) and (B) have been changed to use abbreviations, and the corresponding picture title has also been revised. Please see the ordinate abbreviation of Figure 1(A) and Figure 1(B).

  1. Results: Fig. 2. Correct the ABTS symbols

Response: Thank you for pointing out this mistake. The symbol for “ABTS·+” has been correctly changed to “ABTS·+” in line 131.

  1. Results: Fig. 3 is not so sharp and difficult to read.

Response: Thanks for your kind comment. I believe that clear pictures will be more helpful for editors and readers to understand. The unclear pictures have been replaced. Please see Figure 3. In this way, the relationship between each component (Figure 3A), and the relationship between each indicator (Figure 3B) can be directly reflected.

  1. Discussion: the last sentence is a same as in the abstract.

Response: Thank you for your advice. The same sentence has been modified in the abstract. Please see the abstract section in lines 34-35.

  1. Discussion: Based only on the antioxidant and inhibitory tests and obtained data and not using any in vitro experiments is difficult to do such conclusion. Put as well some references in order to make stronger your general conclusion.

Response: Due to experimental restrictions and limited resources, we did not perform the antioxidant activity in vivo. Therefore, some references have been added to strengthen our general conclusion. Please see lines 330-335. Thank you for pointing this out.

  1. Materials and Methods: How the flowers were prevented from oxidation before freeze drying.

Response: Thank you for pointing this out. After collecting, the inflorescences were immediately transferred into a cool box with ice blocks before transportation to the laboratory and stored at -80°C until it is freeze-dried. This has been briefly added to the method. Please see lines 352-354.

  1. Materials and Methods: As I mentioned above the antioxidant assays were done using different units, therefore it is difficult to compare different results. Check carefully the text.

Response: Due to no single antioxidant assay can be used to determine the antioxidant potential of a sample, we used different assays to evaluate the antioxidant activities of this plant. Although it is difficult to compare the strength of antioxidant activity of different assays, we can compare the antioxidant activities of different samples under the same antioxidant assay by using the equal unit to identify the most active fraction based on the data collected. Our previous study (10.3390/ foods10020315) expressed similar data to this work. We appreciate your advice on this aspect as it is very important for us when it comes to a more in-depth study in our future research. Thank you for your kind comment.

We have tried our best to improve the manuscript. All revisions have been labeled as red, and these changes have not influenced the content and framework of the paper.

We appreciate the Editors/Reviewers’ warm work earnestly and hope that the correction will meet with approval.

Once again, thank you very much for your comments and suggestions.

Yours sincerely,

Assoc. Prof. Xue-Chun Zhang

Southwest Forestry University, KunMing, 650224, China

Tel.: +86-871-63862328;

E-mail address: xuechun_zhang@163.com

Reviewer 2 Report

Line 112 - The correlation value low, to state that DPPH scavenging activity is attributed to phenolics. Please elaborate more (https://doi.org/10.1111/j.1750-3841.2009.01352.x), also please calculate the Specific Antioxidant Activity of the extract as indicated in the article shared.

Support why did you decide to measure all those antioxidant methods in vitro. It is well known that antioxidant activity in vitro is not correlated with the activity observed in vivo. Please add in your manuscript that it is a limitation of the study. 

The method evaluated of enzymes inhibitions are with the crude extract of the plants, without a further processing to obtain metabolites generated during digestion. It would be unlikely that the compounds present in the extract will maintain the chemical from until arriving to the target organ. Although results obtained in the manuscript are valuable, the limitations of the study should be specified. 

Author Response

 Dear Editor in Chief Prof. Freda Sun and Reviewers:

Thank you for giving us the opportunity to revise our manuscript and your valuable comments concerning our manuscript entitled “Phytochemical composition, antioxidant activity, and enzymes inhibitory activities (α-glucosidase, xanthine oxidase, and acetylcholinesterase) of Musella lasiocarpa” (ID: molecules-1292480). The comments are very helpful for improving our manuscript. We have revised the manuscript carefully according to reviewers’ comments point by point. All modifications to the manuscript are highlighted in red for your further comments. The responses to the reviewer’s comments are as follows:

Reviewer: 2

1. Line 112 - The correlation value low, to state that DPPH scavenging activity is attributed to phenolics. Please elaborate more (https://doi.org/10.1111/j.1750-3841.2009.01352.x), also please calculate the Specific Antioxidant Activity of the extract as indicated in the article shared.

Response: Thank you for your valuable comments. We have provided an additional description to explain why the lower correlation value indicated higher scavenging activity of DPPH that was attributed to phenolics. Please see lines 112-113. As suggested (https://doi.org/10.1111/j.1750-3841.2009.01352.x), the samples with higher TPC are with higher specific antioxidant activity, which means TPC contributes to the antioxidant of the sample. However, the DPPH calculation was not expressed by Trolox equivalents in this study. It was difficult to determine the Specific Antioxidant Activity between DPPH and TPC. Thank you for this point. Let us pay more attention to experimental data mining for future research.

2. Support why did you decide to measure all those antioxidant methods in vitro. It is well known that antioxidant activity in vitro is not correlated with the activity observed in vivo. Please add in your manuscript that it is a limitation of the study. 

Response: Thank you for pointing this out. Although there is much evidence regarding the antioxidant effect is not always found in vivo, the in vitro experiments served as a reference for in vivo experiments, and have powerful experimental advantages with convenient, economical, fast and without ethical restrictions. In addition, the limitation of in vitro antioxidant methods has been added to the "3. Discussion" section. Please see lines 330-335.

3. The method evaluated of enzymes inhibitions are with the crude extract of the plants, without a further processing to obtain metabolites generated during digestion. It would be unlikely that the compounds present in the extract will maintain the chemical from until arriving to the target organ. Although results obtained in the manuscript are valuable, the limitations of the study should be specified. 

Response: We fully agree with the reviewer. We have added the limitations of in vitro enzyme inhibition to the “3. discussion” section. Please see lines 330-335.

We have tried our best to improve the manuscript. All revisions have been labeled as red, and these changes have not influenced the content and framework of the paper.

We appreciate the Editors/Reviewers’ warm work earnestly and hope that the correction will meet with approval.

Once again, thank you very much for your comments and suggestions.

Yours sincerely,

Assoc. Prof. Xue-Chun Zhang

Southwest Forestry University, KunMing, 650224, China

Tel.: +86-871-63862328;

E-mail address: xuechun_zhang@163.com

Round 2

Reviewer 1 Report

All comments were addressed, check please once more your text and your style in order to correct minor mistakes

Reviewer 2 Report

The authors properly attended the comment raised by this reviewer. I suggest acceptance of the manuscript in its present form.